# Spray Drying Enzyme-Treated Cellulose Nanofibrils

**DOI:** 10.3390/polym15204086

**Published:** 2023-10-14

**Authors:** Sungjun Hwang, Colleen C. Walker, Donna Johnson, Yousoo Han, Douglas J. Gardner

**Affiliations:** 1Advanced Structures and Composites Center, University of Maine, 35 Flagstaff Road, Orono, ME 04469-5793, USA; douglasg@maine.edu; 2School of Forest Resources, University of Maine, 5755 Nutting Hall, Orono, ME 04469-5755, USA; 3Process Development Center, University of Maine, 5737 Jenness Hall, Orono, ME 04469-5737, USA; colleen.walker@maine.edu (C.C.W.); donna.johnson@maine.edu (D.J.)

**Keywords:** enzyme treatment, cellulose nanofibrils, spray-drying, polypropylene

## Abstract

Enzyme-treated cellulose nanofibrils (CNFs) were produced via a lab-scale mass colloider using bleached kraft pulp (BKP) to evaluate their processability and power requirements during refining and spray-drying operations. To evaluate the energy efficiency in the CNF refining process, the net energy consumption, degree of polymerization (DP), and viscosity were determined. Less energy was consumed to attain a given fines level by using the endoglucanase enzymes. The DP and viscosity were also decreased using the enzymes. The morphological properties of the enzyme-pretreated spray-dried CNF powders (SDCNFs) were measured. Subsequently, the enzyme-pretreated SDCNFs were added to a PP matrix with MAPP as a coupling agent. The mixture was then compounded through a co-rotating twin-screw extruder to determine whether the enzyme treatment of the CNFs affects the mechanical properties of the composites. Compared to earlier studies on enhancing PMCs with SDCNF powders, this research investigates the use of enzyme-pretreated SDCNF powders. It was confirmed that the strength properties of PP increased by adding SDCNFs, and the strength properties were maintained after adding enzyme-pretreated SDCNFs.

## 1. Introduction

The major chemical components of natural fibers are cellulose, lignin, and hemicellulose. Cellulose, the most abundant polymer on earth, is organized into microfibrils of amorphous and strongly hydrogen-bonded crystalline regions (α-cellulose) [1]. Cellulose contains β (1, 4)-linked glucopyranoside monomer units, predominantly located in the secondary cell wall [2]. The three hydroxyl groups on the glucose monomer are attributable to hydrogen bonding among the fibers [3]. The micrometer-sized cellulose can be manufactured into nanometer-sized cellulose by mechanical, chemical, and biological treatments [4]. The generated higher specific surface area with an increased number of hydroxyl groups on each nanofiber leads to an increase in hydrogen bonding, resulting in creating a strong network within the fibers [5].

Cellulose nanofibrils (CNFs) manufactured through mechanical treatment via grinding, refining, and/or homogenization are the most cost-effective production methods and have a very high production rate compared to other methods [6]. Therefore, CNFs are widely used as reinforcing fillers in thermoplastic matrix composites to increase the mechanical properties [7]. However, CNFs have a relatively bigger width and longer fibrils which result in a broad fiber size distribution compared with cellulose nanocrystals (CNCs). Therefore, high energy consumption is required to defibrillate pulp fibers to smaller sizes [8]. Many researchers are actively conducting research to produce CNFs more economically [9].

Conventionally, the bleaching chemicals used in pulping before the refining process eliminate lignin and hemicellulose that act as binding agents between cellulose fibrils, leading to the reduction of refining efficiency [10]. Moreover, TEMPO-oxidation, carboxylation, and sulfonation are widely used to reduce the energy consumption in the refining process; however, those chemical methods are harmful to the environment [11,12,13]. Enzyme pretreatment includes using endoglucanases, cellobiohydrolases (CBHs), and β-glucosidases (BGs), which are well-known environmentally friendly methods for the reduction of refining costs as biodegradable cellulases are neutral and produce no emissions of harmful chemicals [14]. Endoglucanases are the primary enzymatic pretreatment for CNFs among other enzyme treatments. Endoglucanases specifically cleave the cellulose β-1, 4 linkages in the amorphous regions without affecting the crystalline regions. This leads to a reduction in fiber length and an increase in crystallinity, while preserving the mechanical properties of polymer matrix composites (PMCs). Because of their features, many researchers have used endoglucanases as an aid in fiber defibrillation for high efficiency during the production of cellulose nanofibrils [15,16,17,18]. 

Polymer matrix composites (PMCs) are composed of plastic matrices and reinforcement additives [19]. Polypropylene (PP) is a very common commodity thermoplastic, and it has been widely applied in the automotive and packaging industries because of its advantageous properties such as low price, good processability, resistance to weathering, and recyclability, which make PP accepted worldwide with a demand of over 21 million pounds per year [20,21]. Inorganic reinforcing materials including glass, carbon, and aramid fibers are commonly used in the PP matrix to increase its mechanical properties [22,23]. However, compared to conventional inorganic fillers, natural fibers have many significant advantages including biodegradability and relatively high tensile strength. Particularly, a significant enhancement of thermal and mechanical properties occurs with the addition of a small amount of CNFs into the polymer matrix [24]. However, the aqueous slurry CNFs are challenging to use in the manufacturing PMCs industry using the current melt compounding processes [25]. 

Spray drying is a fast, simple, cost-effective, and scalable method, so it is used in various industries including pharmaceutical, food, and chemical manufacturing [26,27]. Furthermore, SDCNFs have been reported to have higher thermal stability and superior crystallinity index than fibers dried by other drying methods including air-drying, oven-drying, freeze-drying, and supercritical-drying [28,29,30,31,32]. Spray-dried cellulose nanofibrils (SDCNFs) have the property of excellent dispersion and distribution in the plastic matrix attributable to their micrometer size with the spherical shape of individual particles [33]. There are typically three different atomizing techniques: a rotary disk atomizer, two-fluid nozzle, and ultrasonic atomizer [34]. Among the three different spray-drying techniques, the pilot-scale rotary disk atomizer offers a larger capacity and improved drying efficiency compared to the other two techniques scaled for laboratory use, attributable to its centrifugal technology that minimizes feed blockage [35]. In a pilot-scale rotary disk atomizer, the feedstock in liquid suspension is transported into the atomizer by a feed pump. The hot air and the centrifugal energy generated by the rotating disk atomizer are delivered to the suspensions. The disintegration of the liquid film results in the formation of droplets by the centrifugal force, and the water in droplets is evaporated, creating dry particles. After the disintegration of liquid film into the formation of droplets, the droplets evaporate, creating dry particles. The resulting particles collide with the surface of the cyclone, leading to a loss of kinetic energy and causing the particles to fall into the collector [36,37,38].

A serious problem of natural fiber use with non-polar polymers is the polar hydroxyl groups on the surface of the fibrils that are incompatible with most plastic matrices [39]. Furthermore, it is believed that the agglomeration of cellulose occurs because of the incompatibility between filler (hydrophilic) and matrices (hydrophobic) as the hydrophilic cellulose fibers can be agglomerated together by the hydrogen bonding among fibrils. Chemical modification on the fibril surfaces can make them hydrophobic, resulting in improved interfacial bonding between PMCs and reinforcing fillers materials [40]. The use of maleic anhydride-grafted polypropylene (MAPP) leads to an increase in the interfacial bonding between fibers and the PP matrix. MAPP can be bonded with the hydroxyl group of cellulose by esterification or hydrogen bonding. At the same time, the PP tail on the MAPP becomes entangled with the melted polypropylene matrix [41,42].

In this research, the net energy consumption, the degree of polymerization (DP), and the low shear viscosity of enzyme- and non-treated CNFs were compared, and they were also compared based on enzyme dosage. The morphological and size analyses of SDCNFs dried from enzyme- and non-treated CNFs were compared. To determine the effect of the enzyme-pretreated SDCNFs on the PP matrix, the enzyme- and non-pretreated SDCNF powders were used as a reinforcing filler in a PP matrix, and the MAPP was used as a coupling agent between two materials, followed by comparing the mechanical properties of two composites. Compared to previous studies that explored the enhancement of PMCs with the incorporation of SDCNF powders, this research evaluates the effects of using SDCNF powders derived from enzyme-treated CNFs instead of non-treated CNFs. It also examines the relationship between the degree of polymerization (DP) and particle production. By reducing the manufacturing cost of CNFs and the subsequent production cost of SDCNFs, we believe this material is viable for use in commodity products and the interior components of the automotive industry.

## 2. Materials and Methods

### 2.1. Enzyme-Pretreated CNFs Production 

The enzyme pretreatments and CNFs production were conducted by the Process Development Center (PDC) at the University of Maine, Orono, ME, USA. CNFs were made using bleached softwood kraft pulp (BSK) (US Patent, US 20170073893A1), after enzyme pretreatment. The enzyme used in this work was pure endoglucanase purchased from FiberCare (Novozymes, Kalundborg, Denmark), and the enzyme activity is 4500 ECU/g. The pretreatments were conducted at two enzyme levels—Low: 0.05% or High: 0.5%—as a percentage of the oven-dried pulp weight. Before addition to the pulp, the enzyme was diluted at a 3:40 ratio with DI water, and the pulp was diluted with the enzyme solution and water to the final treatment consistency of ~4%. The temperature was maintained at 50 °C throughout the treatment by placing the treatment container in a circulatory water bath, and the pulp was constantly mixed with a standing mixer at 1000 rpm. pH was adjusted to 5.5–6.5 by adding 10% H_2_SO_4_ solution as needed. The pulp was held at temperature for a 1 h treatment period. After treatment, the enzyme was denatured by heating the slurry at 90 °C for 25 min. In a representative experiment, 85 g of oven-dried bleached softwood kraft pulp (2219 g slurry at 3.83% consistency) was placed in a container in the water bath. The slurry was stirred until its temperature reached 50 °C. Prior to enzyme addition, the pH of the slurry was adjusted to 6.2. Then, 6.07 g of diluted enzyme solution was added (0.5% enzyme dose) and stirred continuously for one hour. After the treatment time, the mixture was denatured. The enzyme- and non-treated pulps were processed into CNF suspensions by using a mass colloider (Masuku Super, Model MKCA6-2, Tokyo, Japan) at 1800 rpm using ‘fine” plates (MK-E6-46-DD). The pulps were diluted to 1.5% consistency by adding tap water. The pulp was then processed through the mass colloider in a single-pass fashion, repeating until the pulp was fibrillated to low (~50%), medium (~80%), and high (~95%) fines content.

### 2.2. Determination of Fines Level and Energy Consumption

In this study, the fines content (level) of CNF suspensions was reported based on the percentage of under 200 μm length fibers in the total amount of fibers [43]. The fines content was measured via the MorFi Fiber Analyzer (TechPap, Gières, France), and the measurement was made using two cameras that measured the fibers in a 50-micrometer-wide chamber and then delivered the data to the software. The net energy consumption was measured by monitoring the consumption of electricity during the grinding process.

### 2.3. Characterization of Enzyme-Pretreated CNFs

The low shear viscosity of enzyme- and non-treated CNF suspensions were measured via a Brookfield viscometer with spindle #64 at 100 RPM. The degree of polymerization (DP) was estimated based on the intrinsic viscosity according to TAPPI Test Method T237cm-98. *DP* was calculated by the Mark-Houwink-Sakurada equation [44]:$$\left[\eta \right]=K\cdot {DP}^{a}$$
**η* is the intrinsic viscosity; **K* and *a* are the Mark Houwink parameters: *K* = 2.28, *a* = 0.76.

### 2.4. Spray Drying

Cellulose nanofibrils (CNFs) in dry powder form were produced by utilizing a pilot-scale spray dryer. The drying conditions are listed in Table 1. The solids content of all samples was set to 1.5 wt.% before they were spray-dried.

### 2.5. Composite Manufacturing

The following two polymers were used as a matrix and coupling agent for the formulation: polypropylene (PP) (Pro-fax 6525, LyondellBasell, Rotterdam, The Netherlands); maleic anhydride modified homopolymer polypropylene (MAPP) (Polybond 3200, Lawrenceville, GA, USA). SDCNFs-reinforced PP composite was melt-compounded using a co-rotating twin-screw extruder (C. W. Brabender Instruments, South Hackensack, NJ, USA). The extruder process parameters were 180 °C across the heating sections with an extrusion speed of 50 rpm. The composite extrudate passed through a two-nozzle die with a nozzle diameter of 2.7 mm. Cooled extrudates were ground using a granulator (Hellweg MDS 120/150, Hackensack, NJ, USA). A masterbatch compounding process was used in this research to improve the dispersion and distribution of filler within the PP matrix. Table 2 shows the compounding conditions of treated and non-treated SDCNFs-reinforced PP composites. The MAPP ratio was fixed to 5 wt.%, and the PP ratio was changed according to the SDCNF ratios of 5 wt.% and 10 wt.%. For the masterbatching process, the input contents of SDCNFs, MAPP, and PP were 50%, 25%, and 25%, respectively, in the first compounding, followed by adding the neat PP to dilute the masterbatch in the second compounding. The input contents of neat PP and produced masterbatch were 60% and 40%, respectively, as listed in Table 3. The PP composites with 5 wt.% of SDCNFs added were not affected by the masterbatch, so the masterbatch was applied only in the PP composites filled with 10 wt.% of filler content. An injection molder Model #50 “Minijector” with a ram pressure of 2500 psi at 200 °C was used to produce specimens according to ASTM D 638, D 790, and D 256 for tensile, flexural, and IZOD impact tests, respectively.

### 2.6. Morphological Properties of SDCNFs Powders

The SEM images of SDCNFs were obtained via the Hitachi Tabletop Microscope SEM TM 3000 (Hitachi High-Technologies Corporation, Tokyo, Japan). SDCNF powders were placed on the SEM stub covered with carbon tape using a lab scoop. Air flow was then used to secure minor particles to the carbon tape. The set accelerating voltage was 15 kV and various magnifications were adjusted automatically. The particle size distribution was measured via a laser diffractometer Mastersizer 2000 (Malvern, Worcestershire, UK). One gram of SDCNF powders was placed on the tray in the Scirocco 2000 attachment (Malvern, Worcestershire, UK), and the powders were analyzed with a particle refractive index of 1.53 [45]. The aspect ratio and circularity of SDCNFs particles were measured via a Morphologi-G3-ID morphologically directed optical microscope system (Malvern, Worcestershire, UK). The following two equations are Circularity and Aspect Ratio:$$\mathrm{Circularity}=\frac{2\times \sqrt{\pi \times \mathrm{Area}}}{\mathrm{Perimeter}}\;$$
$$\mathrm{Aspect}\;\mathrm{Ratio}=\frac{\mathrm{Width}}{\mathrm{Length}}$$
Perimeter (µm): Actual perimeter of particle; Area (µm^2^): Actual area of a particle in square microns [46].

### 2.7. Mechanical Properties of SDCNFs-Reinforced PP Composite

Tensile strength and MOE were performed according to the ASTM D 638-10 standard [47] and under a displacement control loading with a speed of loading of 5 mm/min. An extensometer was employed to determine the elongation of the specimens. Flexural strength and MOE were performed according to ASTM D790-10 [48] and under a displacement control loading with a speed of 1.27 mm/min. Izod impact strength was measured according to ASTM D256-10 [49] using a Ceast pendulum impact tester (Model Resil 50B). Notching was produced on the impact specimens using a Ceast notch cutting machine.

## 3. Results and Discussion

### 3.1. Effect Enzyme Pretreatment on Energy Consumption

Figure 1 shows the net energy vs. fines curves for the enzyme- and non-treated CNF suspensions. The enzyme-treated pulps required much less energy by up to 64% to reach a 90% fines level than non-treated pulp, and the difference in net energy consumption between the two enzyme dosages was insignificant. The net energy consumption after enzyme treatment was lowered to 70% and 77% compared to the non-treated pulp at the 80% fines level after adding 0.05% and 0.5% enzyme doses, respectively. The reduction rate of energy consumption was the same as 63% at the 90% fines level for two enzyme doses.

Figure 2 and Figure 3 represent the low shear viscosity and the degree of polymerization (DP), respectively. As fines levels increased for both enzyme- and non-treated CNF suspensions, the viscosity increased and the degree of polymerization (DP) decreased, which is consistent with previous research [50,51]. After the grinding process, the increased surface area and aspect ratio of fibers can generate more hydrogen bonding and entanglement between fibers, resulting in strong interfibril interactions. This can restrict suspension flow, leading to increased viscosity [52,53]. DP might be decreased, attributable to the shortened fiber length by the grinding process [54]. In addition, the change rate in viscosity increased, while DP was consistent, as the fine levels increased. In general, longer grinding can further increase the aspect ratio of the fiber because the reduction rate of the fiber width is higher than that of the fiber length. The decreased fiber width with unchanged fiber length increases the aspect ratio, increasing the viscosity change rate and decreasing the change rate of DP [55].

Enzyme-treated 90% fines CNF suspensions had lower DP and viscosity than untreated CNFs by up to 69% and 88%, respectively. DP might be decreased, attributable to the shorter fibers resulting from the cleavage of amorphous regions in cellulose chains by the endoglucanases [56]. In addition, the shorter fiber length by endoglucanases might lead to decreased fiber-fiber interactions, forming a less tight network between fibers resulting in lowered viscosity. Furthermore, the swelling effect resulting from the endoglucanase might make fibers more flexible, leading to separate fiber bundles and reducing fiber-fiber contact sites, resulting in a decrease in the viscosity. A 0.5% dose of endoglucanase had lower viscosity than a 0.05% dose at entire fines levels, attributable to increased binding sites between the fiber surface and the endoglucanase. DP of the 0.5% dose CNFs was higher by 20% compared to a 0.05% dose of endoglucanase at the lower fines levels. However, DP of the 0.5% dose was somewhat lower than that of the 0.05% dose after 80% fines levels, and this is likely attributable to a 0.05% dose of endoglucanase readily removing the exposed cellulose surface chains. In contrast, the rapid removal of cellulose chains might have occurred from the 0.5% dose [57].

It can be concluded that the net energy consumption during a grinding process to reach the targeted fines level decreased with the reduction of viscosity and DP of CNF suspensions attributable to the shortened fiber length and separated fiber bundles by the endoglucanase. In addition, based on the measurement of viscosity and DP, using 0.05% of endoglucanase had enough effect to reduce the net energy consumption during grinding. In this study, only a 0.05% dose of endoglucanase was selected for spray drying and composite manufacture.

### 3.2. Production of SDCNFs Powder

Generally, fine powder spray-dried without a fibrous material might be considered to mean high drying efficiency. In our previous research, good-quality powders without fibrous materials were produced using a 1.5 wt.% CNFs suspension fibrillated by a pilot scale thermo-mechanical refiner. In this research, 1.5 wt.% CNF suspensions were fibrillated through a laboratory-scale mass colloider with a lower fibrillation performance than a pilot-scale thermos-mechanical refiner. All produced SDCNF powders using CNF suspensions fibrillated by the mass colloider included fine powders and fibrous materials after spray drying. This is likely attributable to the fact that CNF suspensions manufactured through a Masuku Super mass colloider are less defibrillated and contain many long fibers, resulting in reduced drying efficiency. The long fibers can be entangled with each other during the drying process, leading to a heavy accumulation of materials on the drying chamber wall and plugging of the spinning disk atomizer holes [58].

As shown in Figure 4, the No. 20 mesh sifting screen (the medium-size U.S. Standard mesh size with an 833 µm nominal sieve opening) was used to sift the SDCNF powders spray-dried from enzyme- and non-treated CNF suspensions and then collect only the fine powders except for the fluffy, fibrous material. Figure 5 represents the SEM images of sifted fine particles and fibrous materials. The spherical-shaped small particles were observed in the collected fine particles. The collected fine powders from enzyme- and non-treated CNF suspensions were used to measure the morphological properties of SDCNF powders and their utilization as the reinforcing material in the polymer matrix.

### 3.3. Effect of Enzyme Pretreatment on Spray Drying

A 90% fines level of CNFs suspension was used for both enzyme- and non-pretreated SDCNFs, and the enzyme-pretreated SDCNFs included 0.05% of endoglucanase. The collected contents of sifted fine powders in enzyme- and non-pretreated SDCNFs were 94% and 84%, respectively. It can be concluded that the spray-drying production efficiency is improved by the enzyme treatment, producing less fibrous material. Figure 6 shows the morphological properties and the particle size distribution of the enzyme- and non-pretreated SDCNFs. For scanning electron microscopy (SEM), particle size distribution (PSD), and Morphologi-G3, the subjected samples were only fine powders separated from fibrous materials; it must be noted that the morphology and size analysis may not perfectly represent the whole samples collected from the spray drying. The CEDs given in the chart are the arithmetical mean values of the particle samples based on the surface area [D3.2] of the samples, and the [D3.2] values were considered as the mean particle size of SDCNF powders in this study [59]. The mean particle sizes of treated and non-treated SDCNFs were 13 µm and 19 µm, respectively. The mean particle size of the 0.05% enzyme-pretreated SDCNFs is slightly smaller than that of the non-pretreated SDCNFs, and this is likely attributable to the endoglucanase reducing the DP of pulp fibers during grinding, leading to more fine particles forming with smaller particle sizes after spray drying. In addition, the fiber bundles might be further separated into individual fibers by a swelling effect resulting from the endoglucanase [60]. The particle size distributions of all samples are well matched with the scanning electron microscopy (SEM) images.

The aspect ratio and circularity of enzyme- and non-enzyme-pretreated SDCNFs were measured via G3-Morphologi (Figure 7). The aspect ratio and circularity values lying between 0 to 1 indicate the particles’ shapes in Morphologi-G3. For example, the closer their value to 1, the closer the shape of the circle, while the closer it is to 0 indicates a more prolonged rod shape [46]. For the aspect ratio in Morphologi-G3, the result value is presented reciprocal to the general result of the aspect ratio. A result value of 1 in Morphologi-G3 means that the length and width of the fibers are the same. Overall, the circularity values were over 85% between 0.8 to 1, and the aspect ratio values were also over 85% in the 0.6 to 1 range, which means that the enzyme- and non-pretreated SDCNFs include spherical-shaped fine particle forms. However, the difference between the two samples was insignificant.

### 3.4. Enzyme-Pretreated SDCNFs Reinforced PP Composites

The enzyme-pretreated SDCNFs were utilized for composite sample manufacturing to evaluate the reinforcing functionality of the CNFs in polymeric matrices. Non-enzyme- pretreated SDCNFs denoted as control SDCNFs in this research were used as a control group to understand the effect of enzyme treatment on the mechanical properties of PP composites. Figure 8 shows the SEM images of 5 wt.% enzyme-treated and non-treated SDCNFs-reinforced PP composites using fractured specimens after the impact test. It can be confirmed that all SDCNFs were embedded in the PP matrix with good dispersion and distribution by the masterbatch system applied using a double compounding process. The tensile strength and tensile MOE of composites increased by adding 5 wt.% and 10 wt.% of enzyme-pretreated and non-pretreated SDCNFs into the PP matrix compared to neat PP; however, there was no significant difference in the increased feeding rate between 5 wt.% and 10 wt.% of fillers added. As shown in Figure 9, the maximum increased rate of tensile strength and tensile MOE were 20% and 45%, respectively. It can be concluded that SDCNFs were well distributed in the PP matrix, and the SDCNFs played the role of reinforcing material properly to the plastic matrix. In addition, the interfacial bonding between hydrophilic SDCNFs and hydrophobic PP matrix is increased by the MAPP coupling agent. Furthermore, it was confirmed that there was no difference in tensile properties even if enzyme-pretreated SDCNFs were added to the PP matrix. The flexural properties showed a similar tendency to tensile properties, and the maximum increase rate of flexural strength and flexural MOE were 7% and 45%, respectively (Figure 10). The impact strength of 5 wt.% SDCNFs-reinforced PP composite increased, and the maximum increase rate of impact strength was approximately 15% (Figure 11). It can be concluded that adding the SDCNFs as a reinforcing material in polymer matrices overcomes the reduction in impact strength attributable to the decreased fiber size with spherical-shaped powders, which reduced the contact site between fibrils and matrices. On the contrary, the impact strength decreased with the addition of 10 wt.% SDCNFs in the PP matrix, and it might be attributed to the result of an excessive amount of filler embedded in the PP matrix. Generally, it is difficult to improve the impact strength using cellulosic materials because of intrinsic long fibrils’ properties resulting in stress concentration that leads to crack initiation in polymer matrices. The further reduction in impact strength occurred when the coupling agent was applied because the strong interfacial adhesion between the fibrils and matrix reduces polymer mobility, and it can prevent fiber pull-outs from the matrix, resulting in a decrease in impact strength [61,62,63]. There was no decrease in mechanical properties, including tensile and flexural properties, and impact strength, even though enzyme treatment was applied to CNFs, which is likely attributable to the result of endoglucanase breaking the amorphous region except the crystalline region of cellulose.

It has been observed that the addition of a small amount of SDCNFs to the polymer matrix in this research significantly enhances mechanical properties, a finding consistent with previous research [33]. According to Peng et al. [33], after adding 6 wt.% of SDCNFs and 2 wt.% of MAPP into the PP matrix, the tensile strength, tensile MOE, flexural strength, and flexural MOE increased by up to 11%, 36%, 7%, and 21%, respectively, compared to the neat PP. In Peng et al., the increase in tensile and flexural properties was lower, and the impact strength was higher than that of this research. The main reason for the different values of mechanical properties might be the different spray-drying techniques used to dry the CNFs suspension. Peng et al. used a lab-scale spray dryer, which utilized a pneumatic two fluid nozzle (TFN) to atomize the droplets of the fluid feed [64], while a pilot-scale rotary atomizer was used in this research. The 4 μm particle size produced by the lab-scale spray dryer was smaller than our powders containing an average of about 13 μm. CNFs dried from a rotary disk atomizer contained bigger particle sizes and more fibrous material with a higher aspect ratio, resulting in better stress transfer between the matrix and the fibers, leading to higher tensile and flexural properties compared to the SDCNFs dried by the lab-scale dryer [65]. In terms of the impact strength, it is believed that the small-sized particles dried by a lab-scale spray dryer prevented more crack initiation, resulting in a higher impact strength [33,66]. Overall, individual SDCNF particles exceeding 10 µm with a higher aspect ratio might be more advantageous for tensile and flexural properties, whereas these larger particles are not as beneficial for impact strength as the finer 4 µm particles.

## 4. Conclusions

In this study, an attempt to reduce the energy consumption used in the CNF grinding process was made through an enzymatic treatment using endoglucanase. It was confirmed that the DP and viscosity of CNFs were reduced by up to 66% and 88%, respectively, and the net energy consumption was lowered by up to 64% at a 90% fines level, attributable to the effect of the enzyme treatment. The enzyme-treated CNF suspensions were successfully dried using a conventional spray dryer, and their sizes are smaller than the control CNFs powder; even the morphologies are similar to each other. The size and morphological differences of the CNFs between the fibrillation methods (super mass collider and disk grinder) need to be studied in the future. The thermoplastic composite samples filled with enzyme-treated SDCNFs showed significant increases in mechanical performance, but no differences from that of composite samples filled with regular SDCNFs. The increase in composite samples was up to 20% and 45% in strength and modulus, respectively. SDCNFs are already known as effective reinforcing materials in PMCs; however, the study of enzyme-treated SDCNFs is limited. Utilizing enzyme treatment on CNF suspensions could reduce SDCNFs manufacturing costs, making them more viable for wider industry applications.

## Figures and Tables

**Figure 1 polymers-15-04086-f001:**
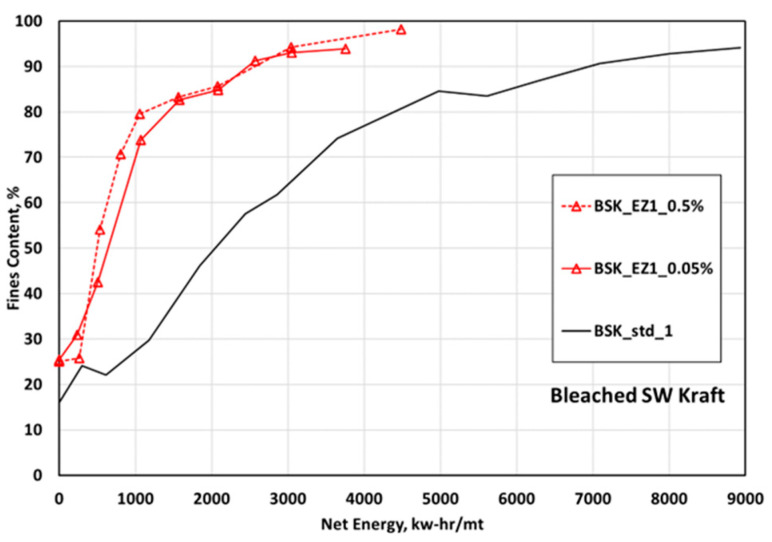
Energy vs. fines for CNFs made from enzyme-pretreated (EZ1) bleached softwood kraft pulp at two doses of the enzyme, 0.5% and 0.05%.

**Figure 2 polymers-15-04086-f002:**
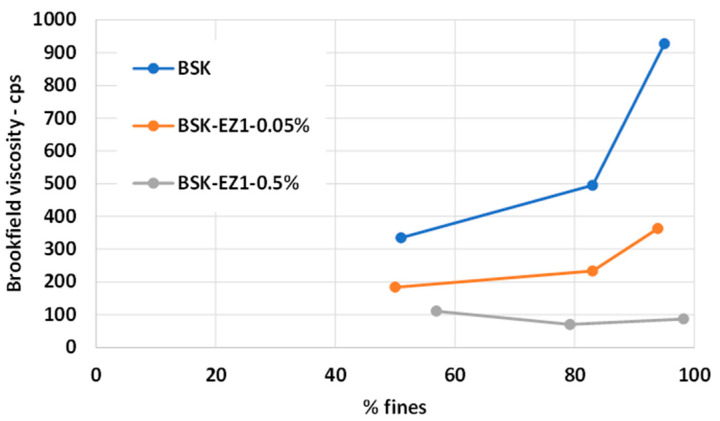
Low shear viscosity vs. fines for CNFs made from enzyme-pretreated (EZ1) bleached softwood kraft pulp at two doses of the enzyme, 0.5% and 0.05%.

**Figure 3 polymers-15-04086-f003:**
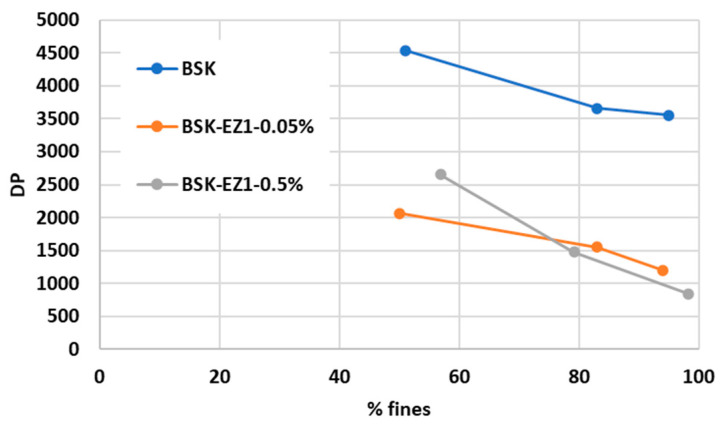
Degree of polymerization vs. fines for CNFs made from enzyme-pretreated (EZ1) bleached softwood kraft pulp at two doses of the enzyme, 0.5% and 0.05%.

**Figure 4 polymers-15-04086-f004:**
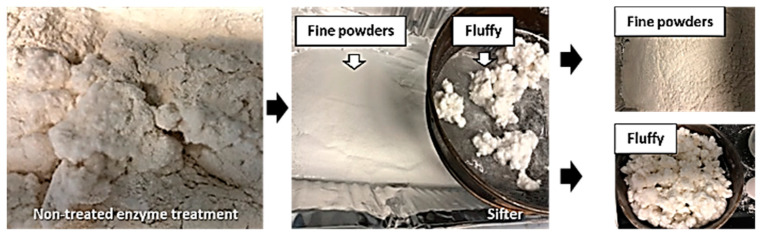
Procedure of sifting SDCNF powders.

**Figure 5 polymers-15-04086-f005:**
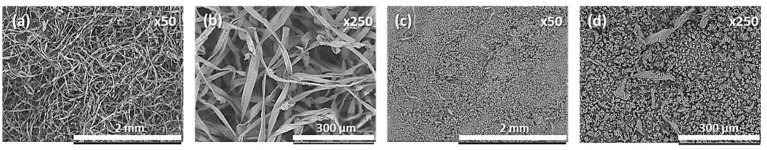
SEM micrographs of sifted fibrous materials (**a**,**b**) and fine powders (**c**,**d**).

**Figure 6 polymers-15-04086-f006:**
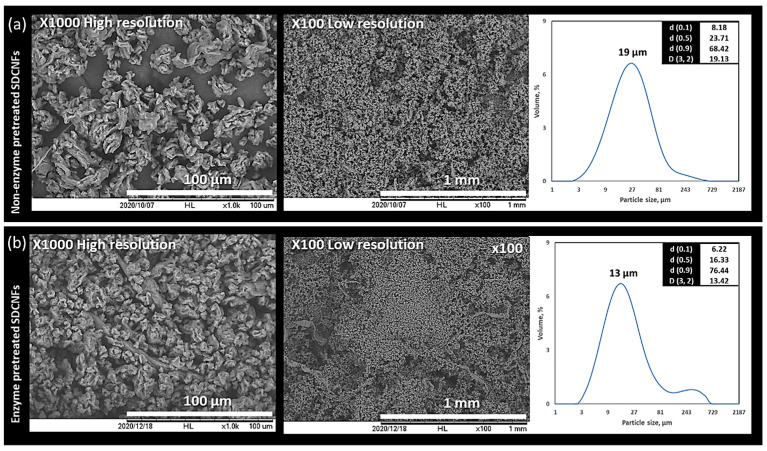
SEM micrographs of non-enzyme-treated (**a**) and enzyme-treated (**b**) cellulose nanofibrils with size distributions and mean CED.

**Figure 7 polymers-15-04086-f007:**
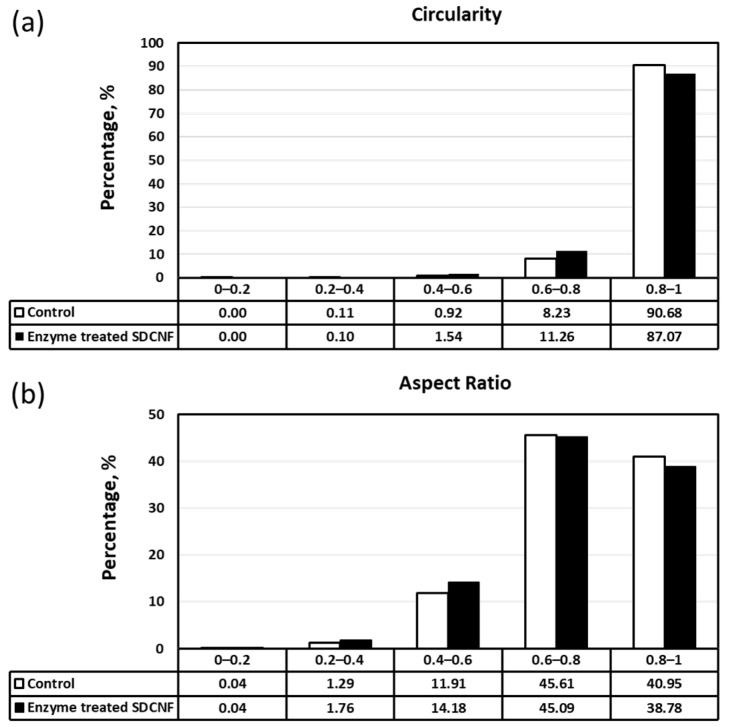
Morphological properties of non-enzyme-treated and enzyme-treated cellulose nanofibrils with circularity (**a**) and aspect ratio (**b**).

**Figure 8 polymers-15-04086-f008:**
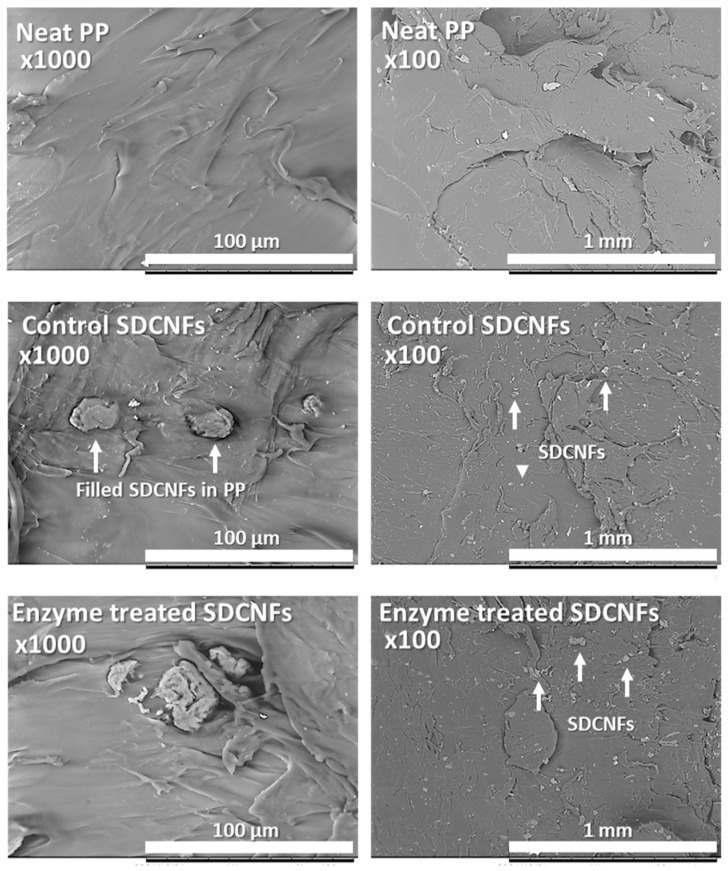
SEM images of fractured neat PP and 5 wt.% reinforced enzyme-treated and non-treated SDCNFs-PP composites.

**Figure 9 polymers-15-04086-f009:**
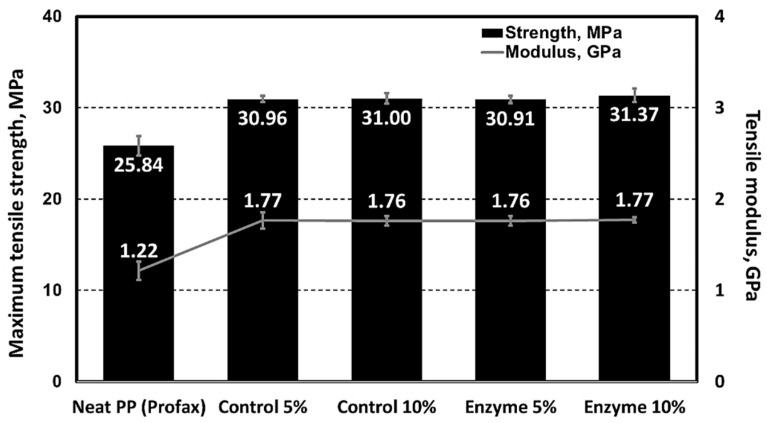
Tensile properties of enzyme-treated and non-treated SDCNFs-reinforced PP composites.

**Figure 10 polymers-15-04086-f010:**
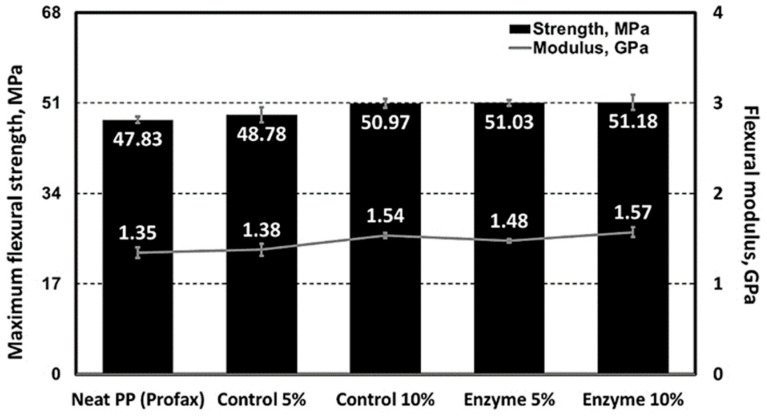
Flexural properties of enzyme-treated and non-treated SDCNFs-reinforced PP composites.

**Figure 11 polymers-15-04086-f011:**
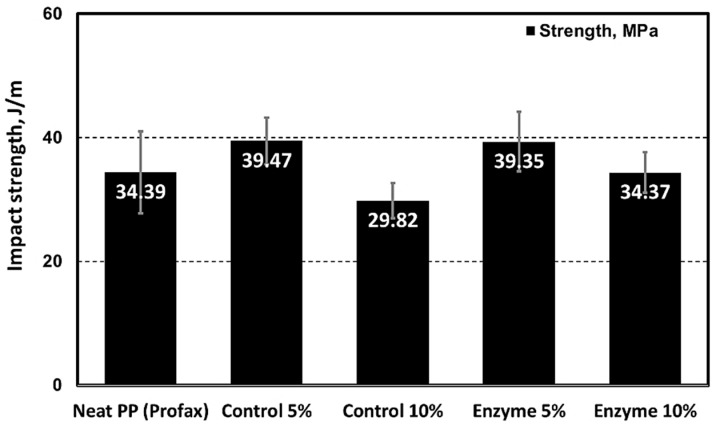
Impact strength of enzyme-treated and non-treated SDCNFs-reinforced PP composites.

**Table 1 polymers-15-04086-t001:** Conditions of spray drying.

Conditions	Inlet Temp, °C	Outlet Temp, °C	Bag House Temp, °C	Spinning Disk, RPM	Feeding Rate, kg/h	Air Fan Speed, %
	248	123	117	30,000	17	85

**Table 2 polymers-15-04086-t002:** Polymers and SDCNFs powder formulation (wt.%).

No.	Composite	PP	SDCNFs	MAPP
1	Neat PP	100	0	0
2	Control 5% (non-enzyme-treated)	90	5	5
3	Control 10% (non-enzyme-treated)	85	10	5
4	Enzyme 5% (enzyme-treated)	90	5	5
5	Enzyme 10% (enzyme-treated)	85	10	5

**Table 3 polymers-15-04086-t003:** Masterbatch formulation (wt.%).

1st Compounding Formulation	2nd Compounding Formulation	SDCNFs	MAPP	PP
SDCNFs50%:MAPP50%:PP25%	Masterbatch40%:PP60%	10	5	85

## Data Availability

All of the material is owned by the authors and/or no permissions are required.

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
