# Peer review of "Spray Drying Enzyme-Treated Cellulose Nanofibrils"

_polymers, 2023, doi:10.3390/polym15204086_

Round 1
Reviewer 1 Report
The experimental article “Spray drying enzyme-treated cellulose nanofibrils” is devoted to a current topic, specifically solving problems when drying a product such as cellulose nanofibrils (CNFs) and the further use of CNFs. In all formal respects, the article corresponds to the Polymers publication. The strengths of the paper include the author's insight into the complexity of CNFs production and the thoroughness of the description of technical solutions to overcome this complexity, as well as the use of enzymes that reduce the degree of polymerization of cellulose under controlled conditions as a tool for solving problems. The manuscript is written in simple language and is small in volume, the figures provided absolutely visualize the processes being studied, but the authors did not formulate the scientific novelty of the presented work either in the abstract or in the conclusion. In addition, some questions and comments arise, a list of which is given below.
Questions and comments:
1. Introduction. Cite published works in 2023 on the topic of the manuscript, since the article will be published in 2023.
2. Fig. 3. Check the number of zeros in the degree of polymerization values.
3. Figure 6. Provide published examples of simultaneous reduction in the degree of cellulose polymerization and reduction in cellulose particle size after exposure to enzymes.
4. Section 3.4. Discuss the reason for the change in the strength of the PP composite when adding CNFs with references to published data and justify it with the results of your own research.
5. Justify the novelty of the results obtained in the manuscript and indicate in addition a short sentence in the abstract and in the conclusion.
Author Response
We sincerely appreciate your comments and advice on our manuscript. We have revised it according to your suggestions and responded to all your comments.
Please see the attachment.

Reviewer 2 Report
In general, the manuscript entitled” Spray drying enzyme-treated cellulose nanofibrils” demonstrated the production of enzyme-treated cellulose nanofibrils (CNFs) via a lab-scale mass-colloider using bleached kraft pulp (BKP), and their processability and power requirements during refining and spray drying operations were evaluated. The research concept is timely and of practical interest, and the conclusion is generally supported by the findings. Therefore, this reviewer would like to recommend the manuscript for publication after addressing the following points.
1. The introduction section should be improved and the novelty should be further highlighted.
2. Detailed information regarding the enzyme samples used in pretreatment should be provided.
3. The specific form of the composite materials prepared by SDCNF should be made clear? Meanwhile, the potential of its application should be further discussed.
4. The advantages of using endoglucanases to produce CNFs with improved properties as compared to other methods should be clarfied. In addition, the present results in the current work should be in comparison to that repored in the references, which will make the paper more readable.
No
Author Response

(The authors gave the same response as above.)

Round 2
Reviewer 2 Report
In general, the revision work is extensive and convincing. Therefore, this reviewer would like to recommend the manuscript for publication.